# The Effects of Attention on the Syllable-Induced Prepulse Inhibition of the Startle Reflex and Cortical EEG Responses against Energetic or Informational Masking in Humans

**DOI:** 10.3390/brainsci12050660

**Published:** 2022-05-18

**Authors:** Xiaoqin Yang, Lei Liu, Pengcheng Yang, Yu Ding, Changming Wang, Liang Li

**Affiliations:** 1Collaborative Innovation Center for Brain Disorders, Laboratory of Brain Disorders, Beijing Institute of Brain Disorders, Capital Medical University Ministry of Science and Technology, Beijing 100069, China; yangxq0204@foxmail.com; 2Key Laboratory on Machine Perception (Ministry of Education), Beijing Key Laboratory of Behavior and Mental Health, School of Psychological and Cognitive Sciences, Peking University, Beijing 100080, China; raylius23@foxmail.com (L.L.); yang0207_psy@163.com (P.Y.); dingyuzero@163.com (Y.D.); 3Division of Sports Science and Physical Education, Tsinghua University, Beijing 100084, China; 4Department of Neurosurgery, Xuanwu Hospital, Capital Medical University, Beijing 100053, China; superwcm@gmail.com

**Keywords:** acoustic startle reflex, prepulse inhibition, attentional modulation, informational masking, energetic masking, event-related potentials, perceptual separation

## Abstract

Prepulse inhibition (PPI) is the reduction in the acoustic startle reflex (ASR) when the startling stimulus (pulse) is preceded by a weaker, non-starting stimulus. This can be enhanced by facilitating selective attention to the prepulse against a noise-masking background. On the other hand, the facilitation of selective attention to a target speech can release the target speech from masking, particularly from speech informational masking. It is not clear whether attentional regulation also affects PPI in this kind of auditory masking. This study used a speech syllable as the prepulse to examine whether the masker type and perceptual spatial attention can affect the PPI or the scalp EEG responses to the prepulse in healthy younger-adult humans, and whether the ERPs evoked by the prepulse can predict the PPI intensity of the ASR. The results showed that the speech masker produced a larger masking effect than the noise masker, and the perceptual spatial separation facilitated selective attention to the prepulse, enhancing both the N1 component of the prepulse syllable and the PPI of the ASR, particularly when the masker was speech. In addition, there was no significant correlation between the PPI and ERPs under any of the conditions, but the perceptual separation-induced PPI enhancement and ERP N1P2 peak-to-peak amplitude enhancement were correlated under the speech-masking condition. Thus, the attention-mediated PPI is useful for differentiating noise energetic masking and speech informational masking, and the perceptual separation-induced release of the prepulse from informational masking is more associated with attention-mediated early cortical unmasking processing than with energetic masking. However, the processes for the PPI of the ASR and the cortical responses to the prepulse are mediated by different neural mechanisms.

## 1. Introduction

The acoustic startle reflex (ASR) is a cross-species, short, quick, and intense defensive reflexive reaction in response to a loud and sudden acoustic stimulus [1,2], which is usually observed by using blink reflexes in humans. The ASR, mediated by the brainstem, can be modulated through a sensorimotor gating process, such as prepulse inhibition (PPI), which is the reduction in the ASR when the startling stimulus is preceded by a weaker, non-starting stimulus (prepulse) [3,4]. PPI can be modulated by attention, healthy human adults exhibit an increase in PPI when instructed to attend to the prepulse stimulus relative to ignoring the prepulse stimulus [5,6,7]. However, the underlying mechanisms in humans are largely unknown.

More interestingly, our recent studies found that the perceptual spatial separation of top-down attention, between the prepulse and the masker, facilitates selective attention to the prepulse, further enhancing PPI in both healthy humans and normal rats [8,9,10,11,12,13]. What is perceptual (perceived) spatial separation? Based on the auditory precedence effect [14,15], when the onset time interval of two different locational sounds with the same content is sufficiently short (1–10 ms), attributes of the delayed sound are perceptually captured by the leading sound, leading to a single fused image whose point of origin is perceived to be around the location of the leading sound [15]. Further, when the target speech image and the masker image were perceived by listeners as coming from two loudspeakers with different locations, this phenomenon is called perceptual spatial separation. By contrast, if the target image and the masker image are perceived as coming from the same loudspeaker, this phenomenon is called perceptual co-location [16].

Perceptual spatial separation can unmask target speech without altering either the signal-to-masker ratio (SMR) or the compactness/diffuseness of sound images [17], it has also been extensively studied in speech masking in reverberant environments. Many studies have found that the recognition of the target speech under the condition of perceived separation was significantly greater than under the condition of perceived co-location, particularly when the masker was speech [17,18]. Auditory masking has two categories: energetic masking and informational masking [19]. Specifically, energy masking is due to the temporal and spectral overlap of the target sound and the masking sound in the auditory periphery. Information masking is caused by masking sounds competing for processing resources with target sounds in the central auditory system [18,20,21]. A noise masker generally generates energetic masking, and a speech masker generates both energetic masking and information masking [18,20]. Therefore, a speech masker can cause a larger masking effect than a noise masker on the target speech. Interestingly, many studies found that perceptual spatial separation, as an unmasking cue, can cause a larger unmasking release of target speech in speech maskers than in noise maskers [18,22]. However, the neural mechanisms underlying the separation effect in humans are largely unknown. Currently, as far as we know, no one has studied PPI under different auditory masking. Whether attentional regulation also affects PPI in this kind of auditory masking, and whether PPI is useful for differentiating noise energetic masking and speech informational masking, are still unknown.

The ERP N1P2 complex, a group of components of early cortical auditory-evoked potentials, can be modulated by attention when a masker is co-presented [23,24]. Interestingly, some studies have used the method of measuring the ERPs of the pulse and the PPI of the ASR simultaneously; they found that the N1 amplitude evoked by the pulse may be a useful indicator for predicting the PPI magnitude [25,26,27]. In other words, the N1 amplitude may represent a brain function that automatically detects sensory changes in the environment and facilitates the execution of appropriate behaviors [26,28,29]. However, some studies have shown that although the ERPs of the pulse (P50, N1, P2, P300) exhibit prepulse inhibition, the magnitude of the inhibition of ERPs and that of the ASR were not correlated [30]. These studies have suggested that the two inhibitory processes are mediated by different neural mechanisms. However, few studies have explored either the cortical responses to the prepulse or the relationship between the prepulse-evoked cortical potentials and the PPI of the ASR. To advance our understanding of the mechanisms underlying the attentional modulations of sensory gating, it is important to further examine whether the prepulse-evoked cortical responses and the PPI of the ASR are functionally related, and even share a common physiological mechanism.

Most of the previous studies on PPI in humans have used meaningless sound stimuli, such as tones or white noise, as the prepulse. In this study, for the first time, a speech syllable, which is more vulnerable to informational speech masking [31], was used as the prepulse against a masking background to explore whether auditory masking types differently affect both the cortical responses to the prepulse and the PPI of the ASR in healthy younger-adult humans, and whether the top-down selective attention to the prepulse, which is facilitated by the perceptual spatial separation between the prepulse and the masker, can further, simultaneously, modulate PPI and cortical ERPs to the prepulse under different auditory masking types. Given the previous findings mentioned above, it was hypothesized that compared to the noise masker, the speech masker may exert a larger masking effect on the early cortical responses to the prepulse, and that both the PPI and the ERP amplitude to the prepulse under the speech-masking condition are more vulnerable to perceptual spatial separation than those under the noise-masking condition.

To date, few studies have measured the attentional modulation of the cortical responses to the prepulse and that of PPI simultaneously. This study simultaneously recorded both the EMGs of startling responses and the EEGs of the prepulse to explore whether the EEGs of the prepulse can be used for predicting the attentional modulation of the PPI of the ASR.

## 2. Material and Methods

### 2.1. Participants

Thirty-five right-handed healthy undergraduate students (20 females, 15 males; 18–30 years; mean age = 21.2 years, SD = 2.5) participated in this study. All the participants were right-handed and had no history of neurological or psychiatric diseases. They had normal (audiometric thresholds < 25 dB HL between 250 and 8000 Hz) and bilaterally balanced hearing (interaural threshold differences at each of the frequencies did not exceed 10 dB). The results of six participants were excluded from data analyses. Three of the participants had no responses to or an extreme habituation to the startling stimulus; two of the participants had frequent spontaneous eye movements, and the EEG data of one of the participants had lost. The results of the remaining 29 healthy participants were used for analyses of both startling response data and EEG data.

This study was conducted according to the principles expressed in the Declaration of Helsinki. All participants gave written informed consent before they participated in this study and were paid a modest stipend for their participation. The experimental procedures were approved by the Committee for Protecting Human and Animal Subjects of the Psychological and Cognitive Sciences at Peking University.

### 2.2. Materials and Apparatus

The prepulse stimuli were two 200-millisecond syllables: /foʊ/ and /faɪ/ (the rise/fall times of two syllables were 20 ms), which were generated from text-to-artificial-speech software, spoken by a female younger-adult. All syllables were presented at 70 dB SPL. Each of the two syllables was used for 50% of trials of each testing session to maintain the participants’ attention and prevent the participant from adapting to the prepulse.

The startling pulse was a 40-millisecond broadband noise burst (0–10 kHz, 104 dB SPL). To study the attention-regulated PPI, there were random 320-millisecond and 280-millisecond intervals between the prepulse and pulse (stimulus onset–onset asynchrony). Each of the two intervals was randomly used for 50% of trials across experiments. Two different intervals were used for removing or minimizing the prediction effects (with deleted content effects and interval effects). In addition, two types of maskers (60 dB SPL) were used: (1) broadband noise (0–10 kHz) and (2) one-talker speech.

During the testing, each session began with a 3-minute acclimation period with one type of masker that was delivered continually from each of the two earphones as the background. The speech masker was a set of linguistically correct but semantically meaningless sentences (e.g., the English translation of one of the nonsense sentences is ‘‘one appreciation could retire his ocean”). Using waveform editing software, each sentence was normalized to equate maximum amplitudes across sentences spoken by a female talker. These masker sentences were repeated in a continuous 5-minute loop. Calibration of stimuli (including the prepulse syllables, speech masker, and noise masker) was based on measuring the root-mean-square (RMS) level and completed by a Larson Davis Audiometer Calibration and Electroacoustic Testing System (Audit and System 831, Larson Davis, New York, NY, USA).

The syllable (prepulse) and the masker were delivered through each of the two earphones with either the right ear leading the left ear by 3 ms or the left ear leading the right ear by 3 ms. As mentioned in the introduction, due to the precedence effect, if a masker in the right ear precedes an identical masker in the left ear by 3 ms, the listener will perceive this masker as coming from the right ear. Furthermore, if the prepulse from the left ear precedes that from the right ear by 3 ms, the listener will perceive this prepulse as coming from the left ear. This perception of prepulse image and masker image as coming from the two ears by listeners is called perceptual separation condition. For the perceptual co-location condition, both the masker and the syllables were perceived as coming from the same ear. The startling pulse was presented simultaneously by both earphones. Participants felt the startling sound image coming from the center of their heads.

### 2.3. Procedures

To examine the effects of (1) masker type and (2) perceptual location relation, two sessions were used to examine all the four possible combinations. Each session (which had only one masker) contained five types of trial (not conditions): (1) 8 prepulse/masker co-location + startling pulse trials (PcP); (2) 8 prepulse/masker separation + startling pulse trials (PsP); (3) 8 startling pulse-alone trials (PA); (4) 152 prepulse/masker co-location trials (PcA, for ERP recordings without the startling pulse); and (5) 152 prepulse/masker separation trials (PsA, for ERP recordings without the startling pulse). PcP, PsP, and PA (baseline) were used to calculate the PPI of the ARS without recording ERP. PcA and PsA were used to calculate ERPs of the prepulse. Finally, the four conditions were PcP in noise masker, PcP in speech masker, PsP in noise masker, and PsP in speech masker.

All the trials were presented in a pseudo-random order in each session, and the order of the first session was counterbalanced. The interval in trials without the startling pulse between prepulse stimuli was set between 1 and 3 s, and the interval in trials with the startling pulse between startling pulses was set between 20 and 25 s (Figure 1).

During the experiment, participants were seated comfortably in a light- and sound-attenuated room. The position of each participant’s head was stabilized by a chin rest. Before the formal testing sessions, each participant received a 3-minute practice session to familiarize them with the sound stimuli and to understand the task. The formal testing (the speech-masker session and the noise-masker session) had a 10-minute break between the sessions. Participants were instructed to attend to sounds presented from the earphones and press a button quickly after a trial if they heard the probe syllable/faɪ/ in one session, and the probe syllable/foʊ/ in another session. To limit eye movements, participants were also asked to watch a cross in the center of a monitor.

### 2.4. Electroencephalogram (EEG) Data Acquisition and Processing

Prepulse-evoked electroencephalogram (EEG) signals were recorded with a 128-channel EEG system (Electrical Geodesics, Inc., Eugene, OR, United States) at a sampling rate of 1000 Hz, digitized using a Net Amps 300 amplifier (10,000-hertz anti-aliasing filter; common-mode rejection 90 dB; input impedance 200 MW). The electrode impedance was kept below 50 kΩ. Data were referenced online to the Cz site and band-pass filtered between 0.03 and 70 Hz.

The raw EEG signals were analyzed and processed using EEGLAB v2019.1. (1) The recordings were filtered at 0.05 and 30 Hz with 24-dB/octave slopes by a Hamming windowed sinc finite impulse response (FIR) filter, which is an embedded function of EEGLAB. The 50-hertz city-power-frequency noise was subjected to notch processing. (2) The reference electrode was changed to the global brain average reference. (3) Artifacts such as body-movement-related-noises, eye movements, and any other high-amplitude noises were excluded by the independent component analysis (ICA). (4) Epoched trials with residual artifacts (mean voltage exceeding ± 70 μV) were excluded from averaging. If the number of artifact trials was more than 25% of the total trials, the participant’s data would be removed from the analysis. In total, the number of remaining trials in each of four conditions across twenty-nine participants as follows: ‘Noise—separation’, 97.9%, ‘Noise—co-location’, 98.1%, ‘Speech—co-location’,94.9%, ‘Speech—separation’, 97.7%.

After processing, the EEG data (epoched trials) were segmented from 100 ms prior to initiation of the sound to 500 ms after the prepulse (PsA and PcA). Data were baseline-corrected according to the 100-millissecond interval before the sound onset. N1P2 waveforms were collected at the site of the frontoparietal area (around Cz electrode) with the average from the five associated electrodes (E106, E112, E13, E7, and Cz) since they were significantly activated on the topographic maps. The latencies and voltage amplitudes of the N1 component were measured as the most negative potential during the 100–210-millisecond interval. The most positive potential occurring after N1 and before 350 ms was considered as the P2 response.

### 2.5. Startle Electromyogram (EMG) Signal Acquisition and Processing

Startle eyeblink EMG responses were measured from the orbicularis oculi muscle with two electrodes (InVivoMetric E220X Ag/AgCl, 4-millimeter recording diameter) that were filled with electrode cream and secured to the face below the right eye. Before the electrode placement, the skin below the right eye was cleaned with a cotton swab soaked in 70% isopropyl alcohol. The electrode wires were connected via a polygraph input box (PIB) to EGI’s Net Amps 300 amplifier with a 1000-hertz sampling rate. This allowed the simultaneous measurement of peripheral nervous system activity (such as EMG) and EEGs.

The raw EMG signal for every trial was scored separately off-line using MATLAB R2018b. The signal was first band-pass-filtered within a 10–500-hertzwindow (with a 50-hertz notch filter), and then epoched using a 1000-millisecond time window (to facilitate the inspection of data quality) with a baseline period of 100 ms preceding the sound onset. Each EMG response was baseline-corrected.

Every trial was examined for signs of spontaneous eye-blinks and possible signs of corrupted EMG signals. Voluntary and spontaneous blinks were excluded from further data analyses using the following exclusion criteria: (1) The latency of the startle response onset (in ms) needed to occur within 20–120 ms after the acoustic startling stimulus; (2) the latency of the startle response peak needed to occur within a window of 0–180 ms after the startling sound onset; (3) outliers of the startle responses were excluded (a deviation of ± 2 SD from the mean). These measures were utilized to calculate the response reactivity of a given trial. None of the subjects had a deletion number of more than three trials in every trial type. In total, the percentages of valid trails in the noise-masking session were: ‘PsP’, 95.3%, ‘PcP’, 95.7%, ‘PA’, 96.7%. The percentages of valid trails in the speech-masking session were: ‘PsP’, 90.8%, ‘PcP’, 92.3%, ‘PA’, 92.3%.

The magnitude of PPI was calculated with the following generally used formula: PPI (%) = (amplitude of the startling sound alone—amplitude of the startling sound preceded by the prepulse)/(amplitude of the startling sound alone) × 100%. The PPI enhancement was calculated with the following formula: PPI enhancement = PPI separation—PPI co-location.

## 3. Results

### 3.1. PPI under Different Conditions

In Figure 2A, a two-way repeated-measures analysis of variance (ANOVA) showed that the PPI under the speech-masking condition was significantly higher than that under the noise-masking condition (*F*(1,28) = 4.456, *p* < 0.05, *ŋ*^2^= 0.137). The results also showed that the PPI produced under the perceived separation condition was significantly higher than that produced under the perceived co-location condition (*F*(1,28) = 12.154, *p* < 0.01, *ŋ*^2^ = 0.303). However, the two-way interaction between the masker type and the perceptual location was not significant (*F*(1,28) = 2.237, *p* = 0.146, *ŋ*^2^ = 0.074).

### 3.2. Control (No Prepulse) Startle Reactivity and Separation-Induced PPI Enhancement (%) under Different Masker Types

To examine whether the difference in PPI was affected by the control startle reactivity, we compared the control startle reactivity on two masker types. As shown in Figure 2B, the paired-sample *t*-tests revealed that there was no significant difference between the control startle reactivities of the noise and the speech masker (*t*
_(a/2)_ = 1.576, *df* = 28, *p* = 0.126). These results showed that the control startle reactivity (baseline) had no effect on the PPI differences under different conditions.

As shown in Figure 2C, the paired-sample *t*-tests revealed that the separation-induced PPI enhancement was larger in the speech masker than in the noise masker (*t*_(a/2)_ = −2.269, *df* = 28, *p* = 0.031).

### 3.3. Cortical Responses under Different Conditions

Figure 3 shows the topographic maps (Figure 3A,B) around the peak latencies of N1 and P2, and grand-mean ERP waveforms (Figure 3C) across the five electrodes (E106, E112, E13, E7, and Cz, for the frontoparietal sites) under the four conditions.

#### 3.3.1. Amplitudes of the N1 Component

As shown in Figure 4A, a two (masker type) by two (perceptual location) two-way repeated-measures analysis of variance (ANOVA) revealed that the N1 amplitude was significantly larger under the noise-masking condition than under the speech-masking condition (*F*(1,28) = 27.325, *p* < 0.001, *ŋ*^2^ = 0.494), and was significantly larger under the perceived separation condition than under the perceived co-location condition (*F*(1,28) = 5.299, *p* < 0.05, *ŋ*^2^ = 0.159). However, the two-way interaction between the masker type and the perceptual location was not significant (*F*(1,28) = 0.517, *p* = 0.478, *ŋ*^2^ = 0.018).

#### 3.3.2. Amplitudes of the P2 Component

As shown in Figure 4B, the two-way repeated-measures analysis of variance (ANOVA) on the P2 amplitudes did not reveal a significant main effects of the masker type (*F*(1,28) = 0.087, *p* = 0.77, *ŋ*^2^ = 0.04), of the perceptual location (*F*(1,28) = 1.14, *p* = 0.295, *ŋ*^2^ = 0.039), or of the interaction between the two factors (*F*(1,28) = 0.087, *p* = 0.77, *ŋ*^2^ = 0.003).

#### 3.3.3. Amplitudes of the N1P2 Complex

As shown in the Figure 4C, the repeated-measures ANOVA showed a significant main effect of masker type (*F*(1,28) = 22.03, *p* < 0.001, *ŋ*^2^ = 0.44), but not of perceptual location (*F*(1,28) = 1.034, *p* = 0.318, *ŋ*^2^ = 0.036), nor did it show a significant interaction between the masker type and the perceptual location (*F*(1,28) = 1.158, *p* = 0.291, *ŋ*^2^ = 0.04).

#### 3.3.4. Latencies of the N1 Component

As shown in Figure 4D, the two-way repeated-measures ANOVA on the N1 latency revealed that the two-way interaction between the masker type and the perceptual location was not significant (*F*(1,28) = 1.238, *p* = 0.275, *ŋ*^2^ = 0.042). However, the N1 latency under the noise-masking condition was significantly shorter than under the speech-masking condition (*F*(1,28) = 11.963, *p* < 0.01, *ŋ*^2^ = 0.299), and the N1 latency under the perceived separation condition was significantly shorter than under the perceived co-location condition (*F*(1,28) = 5.209, *p* < 0.05, *ŋ*^2^ = 0.157).

#### 3.3.5. Latencies of the P2 Component

As shown in Figure 4E, the two-way repeated-measures ANOVA revealed a significant main effect of the masker type (*F*(1,28) = 13.532, *p* < 0.001, *ŋ*^2^ = 0.326), However, the main effect of the perceptual location (*F*(1,28) = 0.742, *p* = 0.396, *ŋ*^2^ = 0.026) and the two-way interaction between the masker type and the perceptual location was not significant (*F*(1,28) = 0.137, *p* = 0.714, *ŋ*^2^ = 0.005).

### 3.4. Correlation between PPI or Startle Reflex and ERP

We examined the correlations between the PPI of the ASR and each ERP amplitude, respectively. However, the Pearson correlation coefficients revealed that there were no significant correlations between the PPI and ERPs under any of the conditions (*p* > 0.05).

### 3.5. Correlation between the PPI Enhancement and the ERP Enhancement

We then examined the correlations between the perceptual-separation-induced PPI enhancement and the perceptual-separation-induced enhancements of each ERP amplitude under two masking conditions (Figure 5). Surprisingly, under the speech-masking condition, the separation-induced PPI enhancement was significantly correlated with the separation-induced N1P2 component enhancement (*r* = 0.398, *p* = 0.032), but not with either the N1 component enhancement or the P2 component enhancement (*p* > 0.05). Under the noise-masking condition, no significant correlation was found between the PPI enhancement and the ERP enhancement (*p* > 0.05).

## 4. Discussion

This study mainly examined whether the top-down selective attention to the prepulse syllable, which is facilitated by the perceptual spatial separation, can modulate the PPI of the ASR and cortical ERPs to the prepulse syllable simultaneously in two masking types. For the first time, we simultaneously recorded both the EMGs of the ASR and the EEGs of the prepulse syllable to investigate whether the cortical EEGs can be used for predicting the PPI of the ASR, and whether the attentional modulation of PPI can be used for differentiating energetic noise masking and informational speech masking.

### 4.1. Effects of Perceptual Separation

In agreement with the results of previous studies, the perceptual separation between the target speech and the masker shifts the masker image away from the target image, facilitates the listener’s selective attention to the target speech, improves recognition of the target speech, or enhances cortical responses to the target speech [10,31,32,33]. In this study, the N1 amplitude under the perceptual separation condition was significantly larger than under the perceptual co-location condition, and the N1 latency under the perceptual separation condition was significantly shorter than under the perceptual co-location condition. Thus, the perceptual spatial separation specifically facilitates the N1 component. The ERP amplitude is usually affected by the magnitude and the synchronization of neural activation, whereas the ERP latency is more related to neural conduction and processing time [34], suggesting that the perceptual spatial separation enhances both the processing depth of the prepulse and the processing speed.

Zhang et al. (2019) have shown that the N1 component is mainly affected by the perceptual features of the acoustic stimuli, while the P2 component is mainly influenced by the listener’s attentional status, suggesting that the P2 component is more related to the top-down modulatory processing [33]. Furthermore, Morales-Muñoz et al. (2016) have shown that the N1 could be related to the mechanisms involved in triggering attention, whereas the P2 could be related to the mechanisms involved in the allocation of attention [35]. Moreover, Annic et al. (2014) have shown that the N1 component is specifically related to stimulus-driven attention, while the P2 component is related to goal-directed attention [36]. However, this study showed that the N1 amplitude, but not the P2 amplitude, was enhanced by the perceptual spatial separation, suggesting that perceptual spatial separation mainly causes improvements in the early cortical representations of the prepulse features, and promotes selective attention to the prepulse.

We also found that under the perceptual separation condition, the PPI was significantly larger than under the perceptual co-location condition. The results were consistent with previous reports in animal and human studies [8,9,10,11,12], suggesting that the perceptual spatial separation between the prepulse and the masker facilitates selective attention to the prepulse, enhancing the early cortical response (the N1 component) to the prepulse in the frontoparietal cortical areas.

### 4.2. Effects of Masker Type

As mentioned in the introduction, a noise masker mainly generates energetic masking, and a speech masker generates both energetic masking and information masking [18,20]. The results of the present study showed that early cortical processing of the prepulse syllable can be used for differentiating the speech informational masker and the noise energetic masker. Under the speech-masking condition, the amplitude of the N1 and that of the N1P2 peak-to-peak complex evoked by the prepulse syllable were significantly smaller, and the latencies of the N1 and P2 components were significantly longer than those under the noise-masking condition. These results may indicate an increase in the processing load under the speech-masking condition because the speech masker not only disturbs the periphery’s prepulse syllable energetically at the perceptual level, but also competes with the prepulse syllable for perceptual resources at the cognitive level. These results are consistent with previous studies [24,31], supporting the view that compared to the noise masker, the speech masker exerts a larger masking effect on the early cortical responses to the target speech.

In this study, PPI and PPI enhancement were investigated under each of the two masking conditions. They were significantly larger under the speech-masking conditions than under the noise-masking condition. At the same time, we did not find a significant control starter reflex difference between noise masking and speech masking. This may support the view that the gating effects on a speech masker include both energy and informational components, and that the perceptual separation mainly reduces the informational masking effect [16,22]. Thus, the two types of masking can be differentiated.

### 4.3. Relations between PPI and Cortical Responses to the Prepulse

Previous studies have shown that the PPI of the ERPs and the PPI of the ASR are not correlated [28,30,37,38], suggesting a functional dissociation between the PPI of the ASR and cortical potential. For example, in healthy humans, a dose-dependent reduction in the PPI of the N1P2 potential, but not of the PPI of the ASR, was observed following clozapine administration [39] while bromocriptine attenuated the PPI of the ASR, and the attenuation was not correlated with the ERP N1P2 complex amplitude [37]. Furthermore, Kedzior et al. (2006) have shown the lack of a significant correlation between the PPI of the ASR and the cortical theta oscillations [30]. These reports suggested that the inhibition of the startle reflex and PPI of ERPs may be mediated by different neural mechanisms. In this study, a correlation between the PPI of the ASR and the ERPs of the prepulse was not found, suggesting that these two processes are also mediated by different neural mechanisms. Indeed, the PPI of the ASR is mediated in the brainstem [40], while the N1 and P2 components are mediated by cortical activity at early cortical processing stages [38]. In other words, the PPI of the ASR cannot be predicted by prepulse-evoked ERPs.

However, in this study, the cortical N1P2-peak-to-peak complex amplitude enhancement induced by the perceptual spatial separation was positively correlated with the PPI enhancement under the speech-masking condition, suggesting that there may be a functional relationship between the attentional modulation (perceptual spatial separation) of cortical syllable processing and the attentional enhancement of the PPI of the ASR under the speech-masking condition. In other words, the separation has an unmaking effect on both the PPI of the ASR and the cortical responses to the speech syllable when the masker is informational speech. Lei et al. (2018) have shown that the N1-amplitude enhancement induced by the perceptual spatial separation was positively correlated with the PPI enhancement induced by the perceptual spatial separation against a noise background when the prepulse was a noise [10]. In the present study, the reason why the positive correlation occurred only under the speech-masking condition may be that the prepulse syllable shared some features with the speech masker, and they might have evoked the same particular group of neurons processing speech signals in the auditory cortex [41,42]. Another reason might be that, as previous studies have shown, under the noise-masking condition, the perceptual spatial separation causes only a small unmasking release of target speech [16,22], but under the speech-masking condition, the perceptual spatial separation can cause a 4~9 dB release of the target speech from informational masking. This indicates that perceptual spatial separation particularly reduces information masking, and its effect on energy masking is very limited. Therefore, the separation effect is more noticeable under the speech-masking condition than under the noise-masking condition. In short, although the PPI of the ASR and the prepulse of the cortical response are mediated by different neural mechanisms, there may be a functional relationship between the attentional modulation of cortical prepulse processing and the attentional enhancement of the PPI of the ASR.

## 5. Conclusions

This study revealed that the perceptual spatial separation facilitates selective attention to the prepulse and enhances not only the PPI of the ASR, but also the early cortical representation of the prepulse, especially in the speech masker. In addition, the ERPs evoked by the prepulse cannot predict the PPI of the ASR, suggesting that the processes for the PPI of the ASR and the cortical responses to the prepulse are mediated by different neural mechanisms.

## Figures and Tables

**Figure 1 brainsci-12-00660-f001:**
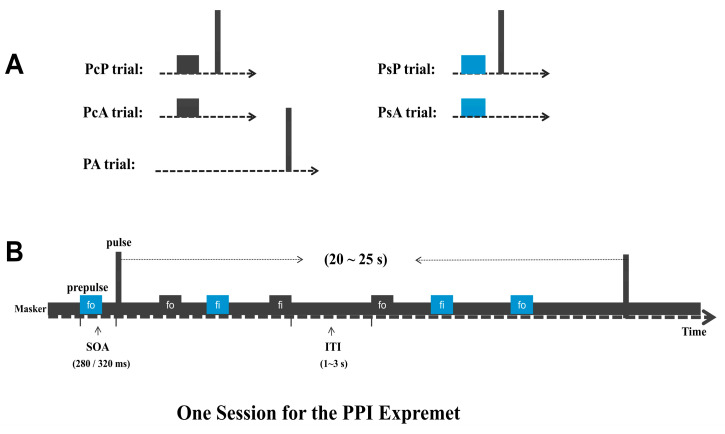
(**A**) Five types of trial. PcP: Trials with prepulse/masker co-location + startling pulse. PsP: Trials with prepulse/masker separation + startling pulse. PcA: Trials with prepulse/masker co-location alone (no startling pulse); PsA: Trials with prepulse/masker separation alone (no startling pulse). PA: Trials with startling pulse alone. (**B**) Flowchart describing the experimental session design. SOA: Stimulus onset asynchrony (onset to onset), ITI: inter-trial interval.

**Figure 2 brainsci-12-00660-f002:**
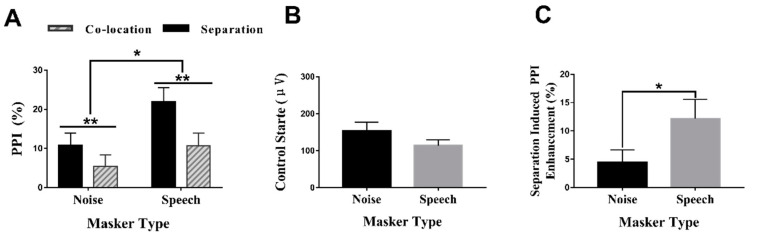
(**A**) Comparisons of PPI (%) across 29 participants under the four conditions. (**B**) Comparisons of control startle reactivity (no prepulse) across 29 participants under two masking conditions. (**C**) Comparisons of separation-induced PPI enhancement across 29 participants under two masking conditions. Formula: separation-induced PPI enhancement = PPI separation—PPI co-location. Data are expressed using mean ± SEM. *, *p* < 0.05, **, *p* < 0.01.

**Figure 3 brainsci-12-00660-f003:**
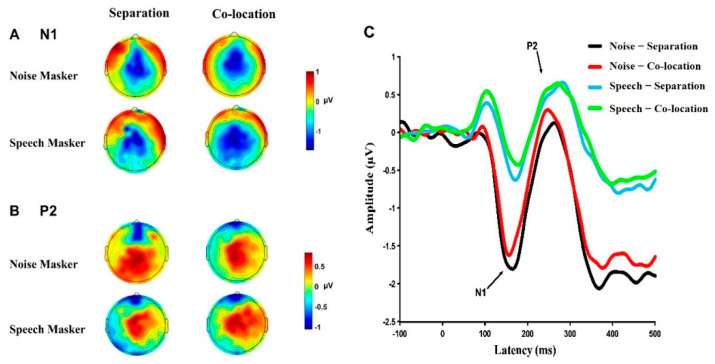
The topographic maps and the grand-mean ERP waveforms evoked by the prepulse. (**A**,**B**) The topographic maps around the peak latencies of N1 and P2 under each of the four masking conditions. (**C**) The grand mean ERP waveforms at the sites near the frontoparietal cortex (averaged across electrodes E106, E112, E13, E7, Cz) under the four masking conditions.

**Figure 4 brainsci-12-00660-f004:**
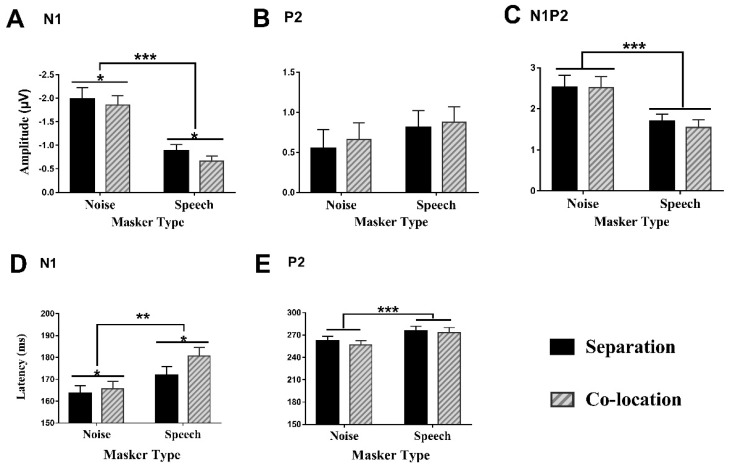
ERPs evoked by the prepulse. (**A**–**C**) Comparisons of the group-mean N1, P2, and N1P2 peak-to-peak amplitudes recorded at the sites near the frontoparietal cortex under the four conditions. (**D**,**E**) The mean latencies of the N1 and P2 components under the four conditions. Data are expressed using mean ± SEM; *, *p* < 0.05; **, *p* < 0.01; ***, *p* < 0.001.

**Figure 5 brainsci-12-00660-f005:**
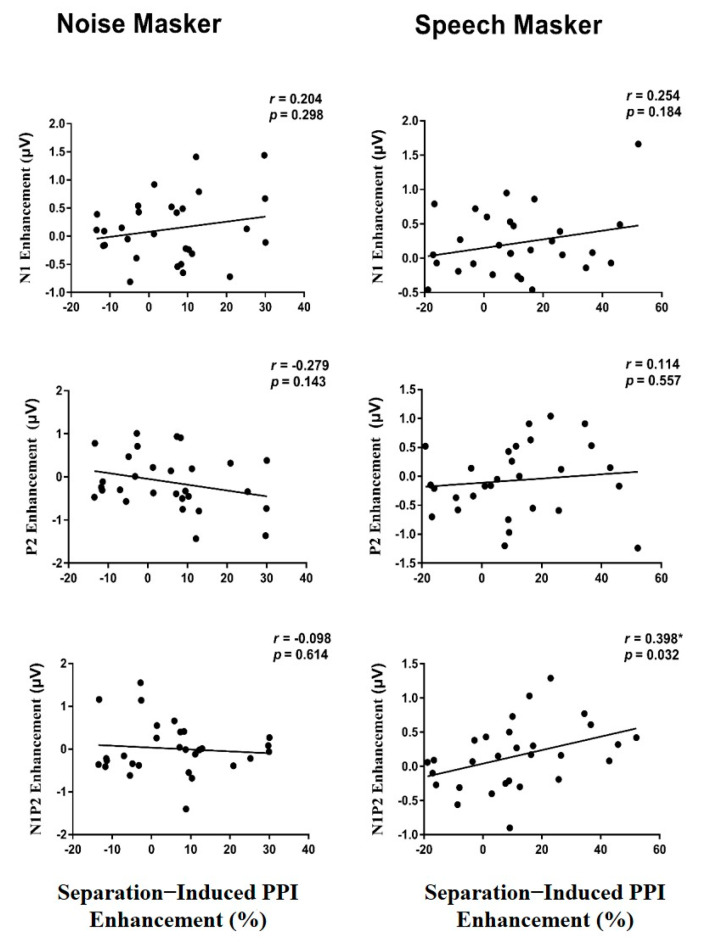
Correlations between the enhancement of the ERP amplitudes of the prepulse and the PPI enhancement under the two masking conditions. The enhancement of the N1P2 peak-to-peak component was significantly correlated with the PPI enhancement under the speech masking condition, but this was not the case for either the N1 enhancement or the P2 enhancement. Note the enhancement variable is the difference score between separation condition and co-location condition. * *p* < 0.05.

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
