# Peer review of "The Effects of Attention on the Syllable-Induced Prepulse Inhibition of the Startle Reflex and Cortical EEG Responses against Energetic or Informational Masking in Humans"

_brainsci, 2022, doi:10.3390/brainsci12050660_

Round 1

Reviewer 1 Report

The manuscript investigates the  Effects of Attention on Syllable-Induced Prepulse inhibition (PPI) of the Startle Reflex. The main neural correlates of PPI lie in the brainstem, However,  that PPI can be top-down modulated by attention (https://doi.org/10.3389/fnhum.2021.649566). I think that the manuscript is important. 

Please find my following comments: 
1- The introduction is very long. There is no sufficient justification of the current study. The authors mentioned that there is clear evidence confirmed the role of attention on PPI. What is the novelty, or rationale?? 

2- Please rewrite the whole manuscript in shorter, and informative way. 

3- Can the study findings recommend for the clinical practice? 

Author Response

请参阅附件。

Reviewer 2 Report

The overall contents of this manuscript are well organized to give a clear overview of this work. The results look promising and are interesting for the readers of Brain Sciences. I have suggested some comments about this study are as the following:

Comments to the Authors:

  1. The introduction section is looking weak. The authors should write this section clearly, like what is the novelty of this work compared to previous literatures.
  2. In method, authors should add a flowchart or system architecture about the EEG signal process.
  3. How authors clean the Raw EEG signals. Does authors performed the independent component analysis (ICA)in EEG lab to remove the noisy from the Raw EEG signals.
  4. Authors should write a new section about the statistical analysis.
  5. In Figure 4 and Figure 5, authors should measure the statistically significant difference between each condition.

Reviewer 3 Report

In this paper, it has been tried to use a new method for repulse inhibition (PPI) when the startling stimulus is preceded by a weaker, non-starting stimulus.

The methods of analyzing and data collection are very good and appropriate. However, there are some points to be improved which could help the reader easily follow the article.

The number of figures is too much whereas some of them could be presented in one table.

Overall, methods and results should be reported more concisely and succinctly.

The font, punctuation, paragraphing and etc. of the whole document should be reconsidered and rechecked.

The citation and references have not followed the rule of the journal.

The whole of the paper does not have coherence and cohesion. This means English editing is needed.

Round 2

Reviewer 1 Report

The manuscript is much improved. However, it stills a ber long. It would be very useful another revision. Please rewrite introduction, discussion, and conclusion in more short and informative way.

Thank you

Reviewer 3 Report

Overall, the manuscript is significantly improved.
